# Correlation between Photocatalytic Properties of ZnO and Generation of Hydrogen Peroxide—Impact of Composite ZnO/TiO₂ Rutile and Anatase

Nouha Mediouni [1,2], Frederic Dappozze [1], Lhoussain Khrouz [3], Stephane Parola [3], Abdesslem Ben Haj Amara [2], Hafsia Ben Rhaiem [2], Nicole Jaffrezic-Renault [4], Philippe Namour [5] and Chantal Guillard [1,*]

1   Institut De Recherche Sur La Catalyse Et l'Environnement De Lyon (IRCELYON), CNRS, UMR 5256, Université Lyon 1, F-69626 Villeurbanne, France
2   Laboratoire Ressource Matériaux et Ecosystème, Faculté des Sciences de Bizerte, Université de Carthage, Zarzouna 7021, Tunisia
3   Laboratoire de Chimie, Ecole Normale Supérieure de Lyon, CNRS, UMR 5182, Université Lyon 1, F-69364 Lyon, France
4   Institute of Analytical Sciences, University of Lyon, F-69100 Villeurbanne, France
5   INRAE, UR RiverLy, Centre de Lyon-Grenoble Auvergne-Rhône-Alpes, F-69625 Villeurbanne, France
*   Correspondence: chantal.guillard@ircelyon.univ-lyon1.fr; Tel.: +33-0472445316

**Abstract:** The generation of hydrogen peroxide on commercial and synthesized ZnO from different precursors was studied using two model molecules, formic acid (FA) and phenol (Ph), as well as phenolic intermediates, hydroquinone (HQ), benzoquinone (BQ), and catechol (CAT). The samples were characterized using X-ray Diffraction (XRD), Transmission Electronic Microscopy (TEM), RAMAN, and Electron Paramagnetic Resonance (EPR) before evaluating their photocatalytic properties. We found that the improved efficiency is accompanied by a high level of $H_2O_2$ production, fewer oxygen vacancies, and that the number of moles of $H_2O_2$ formed per number of carbon atoms removed is similar to the degradation of FA and Ph with a factor of 1. Moreover, a comparative study on the formation of $H_2O_2$ was carried out in the presence of $TiO_2$ rutile and $TiO_2$ anatase, with commercial ZnO. Our results exhibit the impact of the presence of $TiO_2$ on the decomposition of hydrogen peroxide and the formation of phenolic intermediates, which are much lower than those of ZnO only, which is in agreement with the formation of hydroxyl radicals $°OH$ and superoxide $O_2^{°-}$ degrading significantly hydroquinone (HQ), benzoquinone (BQ), and cathecol (CAT).

**Keywords:** photocatalysis; $H_2O_2$; ZnO; oxygen vacancies; ZnO/TiO₂

## 1. Introduction

Water pollution is becoming a universal threat that leads to environmental degradation and poses a risk to humans and the environment worldwide. Advanced Oxidation Processes (AOP) are emerging as an alternative and promising technology for wastewater pollution control [1,2], they are based on the formation of hydroxyl radicals ($°OH$) exhibiting a higher oxidizing power able to break down organic molecules into degradable molecules or mineral compounds [3,4]. Among different methods, heterogeneous photocatalysis was the most promising process for water problems [5–8]. Under UV light, the semiconductor generates electron/hole pairs ($e^-/h^+$) leading to highly Reactive Oxygen Species (ROS): the holes ($h^+$) react with the electron donors adsorbed on the surface to form hydroxyl radicals ($°OH$), the oxygen is reduced by electrons, producing superoxide ($O_2^°$) radicals, and it is in the presence of protons becoming hydro-peroxide ($HOO°$) radicals [9,10]. Many studies focused on understanding the generation of hydroxyl radicals ($°OH$) via the oxidation route by trapped holes, assumed to be the most important factor for oxidation processes in photocatalysis [11–14]. However, hydrogen peroxide ($H_2O_2$) has evolved for years as an interesting reservoir for the generation of hydroxyl radicals

and a key parameter for understanding photocatalytic mechanisms [15–17]. Under UV irradiation, $H_2O_2$ can be formed, either directly by the reaction of photo-generated electrons with adsorbed molecular oxygen or by the reduction in the superoxide radicals ($O_2^{\circ-}$), and the hydro-peroxide ($HOO^\circ$) already present in the system can be further reduced and transformed into $H_2O_2$ [18,19]. Depending on the photocatalyst, $H_2O_2$ can be continuously generated, as is the case in the presence of ZnO [20,21], or decomposed and generally cannot be detected in the solution as in the case of $TiO_2$ [22–24]. This difference was attributed to the complexation of $H_2O_2$ on $TiO_2$ [25], which is not as in the case with zinc oxide. Moreover, the structure of $TiO_2$ also plays a role in the rate of disappearance but also in the nature of reactive oxygen species generated. Sahel et al. [26] showed that hydrogen peroxide was more rapidly decomposed in the rutile phase compared to the anatase phase. Moreover, Hirakawa et al. [27] found that $O_2^{\circ-}$ species were formed during the decomposition of $H_2O_2$ in the anatase phase while hydroxyl radicals ($HO^\circ$) were preferentially generated in the rutile phase. On ZnO, Jang et al. [28] reported that the generation of $H_2O_2$ depends on the morphology of the crystal and that the most active ZnO generates the highest amount of $H_2O_2$. Furthermore, Domenech et al. [29] reported that the type of electron donors, such as phenol, oxalate, and 2,4-dinitrophenoxyaceticacid have a significant effect on the production of $H_2O_2$ in agreement with the role of the hole scavengers necessary for the formation of $H_2O_2$ [30]. Mrowetz et al. [31] studied the formation of $H_2O_2$ during the degradation of formic acid (FA) and benzoic acid (BA) on $TiO_2$ and ZnO and confirmed the higher formation of $H_2O_2$ on ZnO. They suggest that the more important reactivity of $TiO_2$ on the degradation of formic acid, while ZnO is the best on the degradation of benzoic acid, could be linked to the decomposition of $H_2O_2$ in the presence of FA, whereas in the presence of BA the formation of phenolic intermediates adsorbed on $TiO_2$ avoids the adsorption of $H_2O_2$ and its decomposition. To better understand the role of $H_2O_2$, several works studied the impact of the addition of $H_2O_2$ on the photocatalytic properties of $TiO_2$ and ZnO [32–38]. Whereas Khodja et al. [33], Poulios et al. [35], Barakat et al. [37], and Domingues et al. [38] showed a beneficial impact, Pichat et al. [32], Dionysiou et al. [36], and Daneshvar et al. [34] reported that this benefit depended on the ratio between the $H_2O_2$ concentration and pollutant concentration. In the presence of a high concentration of $H_2O_2$, the $^\circ OH$ radical can react with $H_2O_2$ and have a harmful effect. In summary, an understanding of the photocatalytic behavior of hydrogen peroxide is still far from being reached. In most previous studies, researchers considered the formation process of hydrogen peroxide to clarify reaction mechanisms that are occurring, tried to explain why it is detected and why not when using semiconductors, and proposed some possible roads of its generation. However, to our knowledge, the correlation between the disappearance of pollutants and the formation of spontaneous $H_2O_2$ on different ZnO as well as the impact of $TiO_2$ composite on its formation has not been established.

The originality of our work was to investigate if a correlation could be established between the photocatalytic activity of the different ZnO and the $H_2O_2$ formation independent of the organic compound used. Furthermore, to investigate the impact of the presence of $TiO_2$, rutile, and anatase mechanically mixed with ZnO on the hydrogen peroxide formation and on the initial intermediate formed during the photocatalytic degradation of phenol to check if the known $TiO_2$ rutile generated $^\circ OH$ radicals by $H_2O_2$ decomposition was the best compared to known $TiO_2$ anatase generated $O_2^\circ$ that is less efficient.

## 2. Results

### 2.1. Catalysts Characterization

The impact of the precursor was first determined in relation to the morphology of the ZnO obtained (Figure 1). As shown, in all the cases after calcination at 400 °C, a spherical morphology is obtained with well-dispersed particles and an average particle size between ~14 ± 2 nm and ~27 ± 2 nm. However, commercial ZnO presents larger particles (37 nm) with non-homogeneous morphologies.

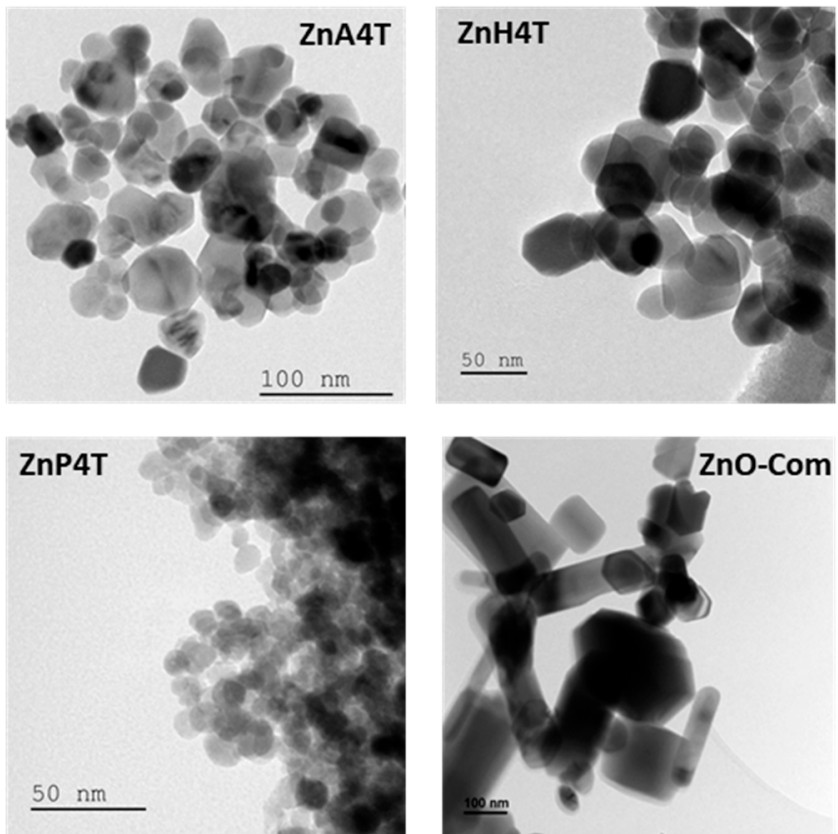

**Figure 1.** TEM images of ZnO nanoparticles and commercial ZnO as reference.

The structural properties of the as-prepared ZnO samples were characterized using X-ray Diffraction. As shown in Figure S1, the crystalline phase of the two precursors indicates the formation of $Zn(OH)_2$ and $ZnO_2$ without any impurities. The XRD spectra of all ZnO samples obtained from different precursors, together with commercial ZnO as reference are presented in Figure 2a. In all the cases and at the same calcination temperature, the samples present the same phase composition with characteristic planes (100), (002), and (101), corresponding to a pure phase Hexagonal Wurtzite of ZnO (JCPDS 36-1451). The crystallite size of the resulting ZnO from zinc acetate, zinc nitrate, and zinc peroxide was calculated using Scherer's equation [39] and values of 19 nm, 23 nm, and 14 nm, respectively, were determined. For commercial ZnO, bigger crystallites (size: 40 nm) are obtained in good agreement with the morphological characteristics shown in Figure 1. The corresponding surface areas and the crystallite size of different samples are presented in Table 1. The change of precursor does not induce a change in the morphology remaining spherical of the resulting samples. However, the use of different types of precursors affects the defect states of ZnO samples, which can be observed using electron paramagnetic resonance (EPR) and Raman (Figure 2b,c).

**Table 1.** Average crystallite size, BET surface area, and concentration of °OH radical measured using EPR for the different ZnO nanoparticles.

|  | ZnA4T | ZnH4T | ZnP4T | ZnO-Com |
|---|---|---|---|---|
| Crystallite size (nm) | 19 | 23 | 14 | 40 |
| BET ($m^2/g$) | 34 | 17 | 17 | 17 |
| $C_{°OH}$ (mM) | 11.2 | 8.2 | 6.1 | 11.1 |

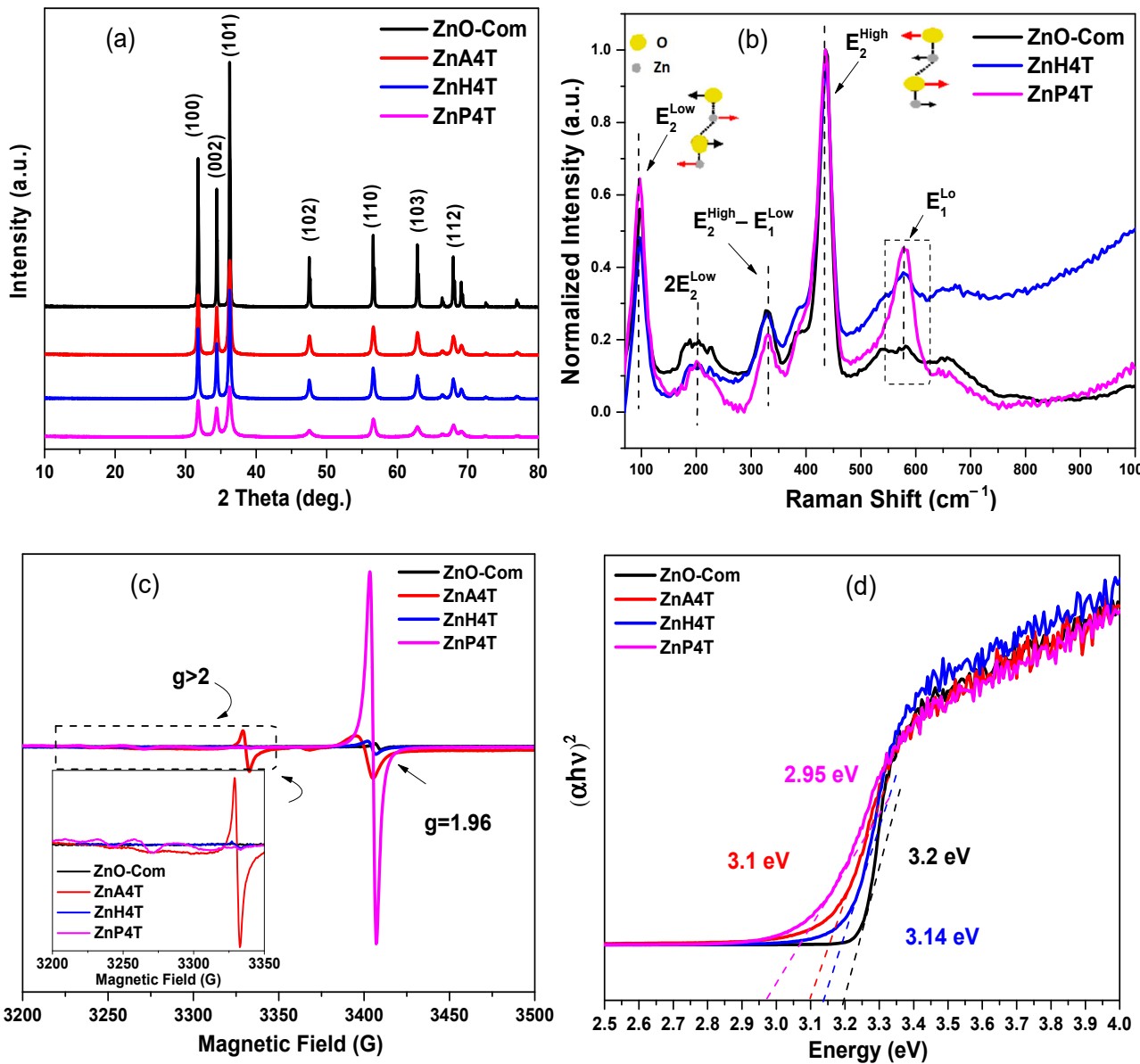

**Figure 2.** (**a**) X-ray diffraction patterns, (**b**) Normalized Raman spectra, (**c**) Electron Paramagnetic Resonance (EPR), and (**d**) Energy band gap of as-prepared ZnO samples including commercial ZnO as reference.

Figure 2b shows the normalized spectra of commercial ZnO and homemade ZnO, except for ZnO prepared from zinc acetate that exhibits an excessive fluorescence. As shown, the characteristic peaks located at 98 cm$^{-1}$, 202 cm$^{-1}$, 330 cm$^{-1}$, and 437 cm$^{-1}$, represent typical vibration modes of ZnO Hexagonal Wurtzite, which correspond to $E_2^{low}$, $2E_2^{low}$, $E_2^{high} - E_2^{low}$, and $E_2^{high}$ modes, are identified in all the cases. On the other hand, an additional vibration mode at 580 cm$^{-1}$ was also observed. This peak is recorded to ($E_1^{Lo}$) mode and is related to the oxygen vacancies appearing in the structure of ZnO. The difference in defect contribution, represented by the $E_1^{Lo}$ mode, is observed between catalysts. The highest level exhibiting the dominance of oxygen defects in the structure is recorded with the sample prepared from zinc peroxide. To characterize defects and investigate the effects generated by using different types of precursors, Electron Paramagnetic Resonance (EPR) was employed (Figure 2c).

In all cases and for all samples, there is a first-order EPR signal at g = 1.96. As reported, this resonance was explained by the existence of oxygen vacancies in the structure

of ZnO [40,41]. As shown, an additional paramagnetic resonance is detected at g > 2 attributed to zinc vacancies [42–45], with a different distorted environment as previously observed in the literature [46].

Figure 2d shows the bandgap energy of the ZnO samples, determined using the Kubelka–Munk method by comparing all the samples, a difference in their optical properties is observed. The commercial ZnO exhibits a bandgap value of 3.2 eV, compatible with the value indicated in the literature [47]. However, with the use of different precursors, the bandgap energy of the samples decreases to 3.15 eV (ZnH4T), 3.1 eV (ZnA4T), and 2.9 eV (ZnP4T), respectively. Combined with the Raman characterization, these changes are consistent with the increase of the concentration of oxygen vacancies generated in the ZnO structure, which is more important with zinc peroxide as a precursor and therefore exhibits the lowest energy of bandgap.

The XRD of the commercial $TiO_2$ P25, A100, and R100 are given in Figure S2. As shown, pure anatase and rutile samples are, respectively, present in A100 and R100, while for P25 a mixture of the anatase and rutile phase is well observed.

The optical properties of the different samples of $TiO_2$, including the mixture of ZnO-anatase and ZnO-rutile together with commercial ZnO, are provided in Figure S3. The bandgap energy of $TiO_2$-based materials is situated between ~3 eV and ~3.2 eV, in the same range with a bandgap value of ZnO (~3.2 eV). As expected, a smaller bandgap is observed with the rutile phase (~3.01 eV). On the other hand, the composites $ZnO/TiO_2$ (anatase and rutile) do not significantly affect the optical properties of the obtained samples. A slight modification is observed by combining with the rutile phase (smaller bandgap ~3.1 eV), which is in agreement with the result of the rutile only, which is close to the visible range.

### 2.2. Characterization of Radicals Intermediates Using EPR

The EPR spectra of ZnO-Com, ZnA4T, ZnH4T, and ZnP4T obtained under UV irradiation for 10 min in aqueous solutions and in the presence of DMPO are presented in Figure 3. For all precursors, a strong signal (marked by @) attributed to $^\circ$OH is observed (92–100% of the total signal). This species (aN = 14.89 G and aH = 14.89 G) is assigned using a simulation (Figure S4) [48–52]. Moreover, except for ZnP4T, an additional species is detected at aN = 15.6 G and aH = 19 G (marked by *). This signal is a C-centered (or $CO_2^\circ$) species and could be attributed to the presence of an organic impurity at the surface of ZnO [48,51,52]. After the addition of formic acid (FA) or phenol (Ph) as model pollutants and after UV irradiation, the evolution of the $^\circ$OH EPR signal is different between FA and Ph (Figure S5). In the case of formic acid, the $^\circ$OH EPR signal completely quenches and the organic species generated are the same for all photocatalysts suggesting that the mechanism of the decomposition reaction is similar (Figure S5a). However, in the case of phenol, a difference is observed between the photocatalysts indicating a difference in the reaction process (Figure S5b). The EPR signal of organic species is totally absent with ZnP4T, while it exhibits a significant signal of $^\circ$OH species in the presence of Ph. We also note the absence of the $^\circ$OH signal in the case of commercial ZnO, while it is always present with a percentage of 30% in the case of ZnA4T (Figure S6). The organic species detected in the presence of phenol (aN = 15.61 G, aH = 18.77 G) and (aN = 15.5 G, aH = 23 G) are compatible with C-(O) (or Ph-(OH)) and Phenyl radicals, respectively (Figure S6).

An estimation of $^\circ$OH radicals has been performed using the calibration curve provided in Figure S7. The values were provided in Table 1.

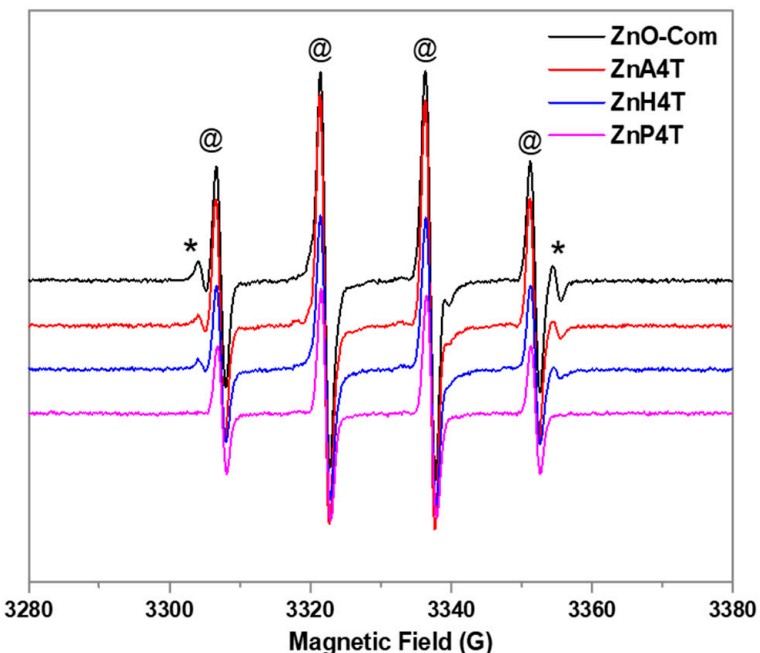

**Figure 3.** EPR spectra of ZnO elaborated from different precursors including commercial ZnO under UV light and in the presence of DMPO.

*2.3. Formation of $H_2O_2$ during the Photocatalytic Reaction of HCOOH and Phenol*

2.3.1. Comparison of the Formation of $H_2O_2$ from FA and Ph in the Presence of ZnO-Com

In the absence of a pollutant, $H_2O_2$ is formed for about 1 h and then it reaches a pseudo plateau, whereas, in presence of a pollutant, the formation of $H_2O_2$ is more important (Figure 4).

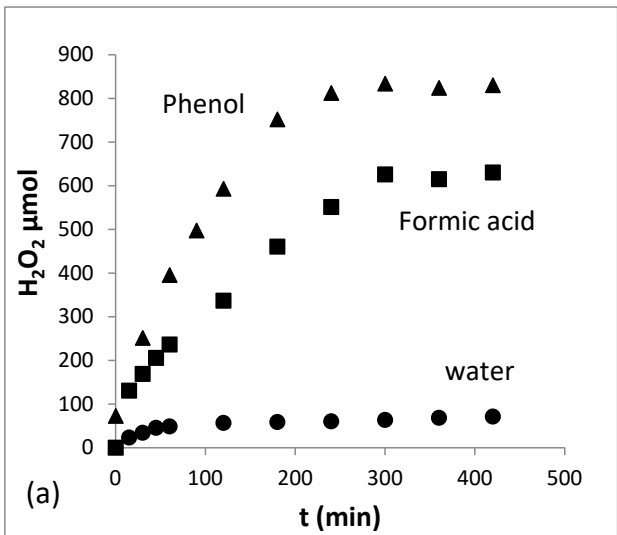

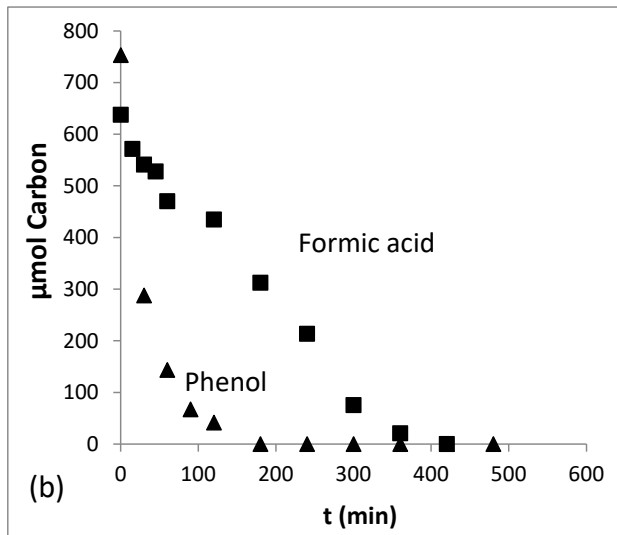

**Figure 4.** (**a**) Formation of $H_2O_2$ in the presence of commercial ZnO and irradiation of an aqueous solution with and without pollutant and (**b**) disappearance of phenol and formic acid, expressed in µmol of carbon, as a function of irradiation time.

The formation of $H_2O_2$ in water, with no added hole scavengers, is explained considering the reaction of $(e^-, h^+)$ pairs formed with oxygen and water following Equations (1)–(6):

$$O_2 + e^- \rightarrow O_2{}^{\circ-} \tag{1}$$

$$H_2O + h^+ \rightarrow {}^\circ OH + H^+ \tag{2}$$

$$H^+ + O_2{}^{\circ -} \rightarrow (HO_2{}^\circ)_s \tag{3}$$

$$(HO_2{}^\circ)_s + e^- \rightarrow (HO_2{}^-)_s \tag{4}$$

$$(HO_2{}^-)_s + H^+ \rightarrow H_2O_2 \tag{5}$$

However, in parallel to the formation of $H_2O_2$, the hydroxyl radicals are formed which, in the absence of a pollutant, can react with $H_2O_2$ decreasing its formation explaining the pseudo plateau observed in Figure 4a.

$$H_2O_2 + {}^\circ OH \rightarrow H_2O + HO_2{}^\circ \tag{6}$$

In the presence of pollutants, the formation of $H_2O_2$ is more important. This behavior has already been observed by some authors [29,30]. As suggested, the photocatalytic generation mechanism of hydrogen peroxide occurs mainly via the reduction in adsorbed oxygen. Accordingly, the increase of $H_2O_2$ in the presence of pollutants is explained by a decrease of charged recombination due to the reaction of pollutants with $^\circ OH$ or with $h^+$ favoring the reduction of $O_2$ into $O_2{}^{\circ -}$. From Figure 4b, it can be noticed that a higher disappearance of phenol than that of HCOOH, led to a higher formation of $H_2O_2$. Considering the higher initial concentration of carbon in phenol solution, 1276 µmol/L against 1086 µmol/L for FA, the final amount of $H_2O_2$ formed seems to be proportional to the initial concentration of carbon present in the solution. This result agrees with the formation of $H_2O_2$ obtained for two concentrations of phenol 20 ppm and 30 ppm corresponding to 766 µmol and 1149 µmol of carbon in the 600 mL of solution, respectively (Figure 5).

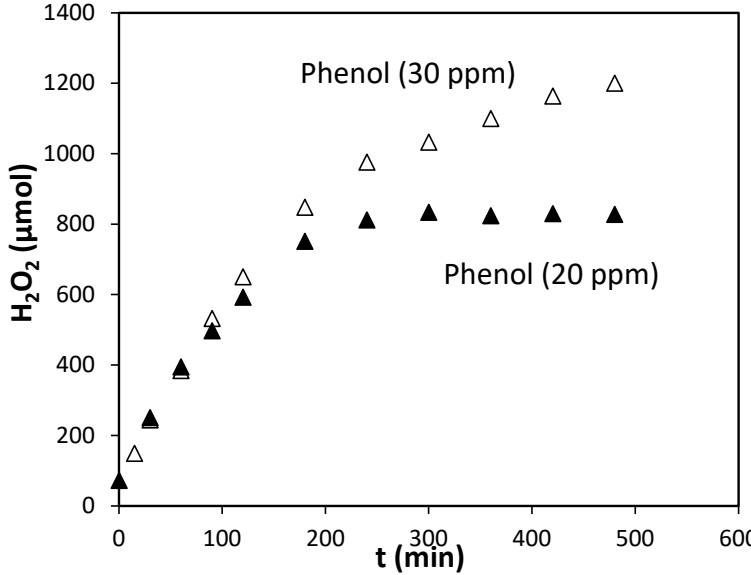

**Figure 5.** Effect of the phenol concentration during the evolution of hydrogen peroxide using commercial ZnO.

Moreover, it can be seen from Figure 6 that decreasing the ZnO concentration from 1 g/L to 0.2 g/L has no effect on the generation of $H_2O_2$ and the photocatalytic degradation of phenol. This result shows that 0.2 g/L of catalyst was enough to absorb all the photons and reach about 95% of efficiency after only 180 min of irradiation.

In summary, a modification of the ZnO concentration from 0.2 g/L to 1g/L does not lead to any change in the efficiency of the degradation and the generation of $H_2O_2$. Moreover, the final concentration of $H_2O_2$ formed seems independent of the nature of the pollutant and only depends on the amount of initial carbon present in the solution. Thus, to better understand the mechanism of $H_2O_2$ formation in the presence of these two pollutants, the amount of $H_2O_2$ formed in presence of commercial ZnO during UV

irradiation is reported as a function of the number of carbons removed from HCOOH or phenol (Figure 7).

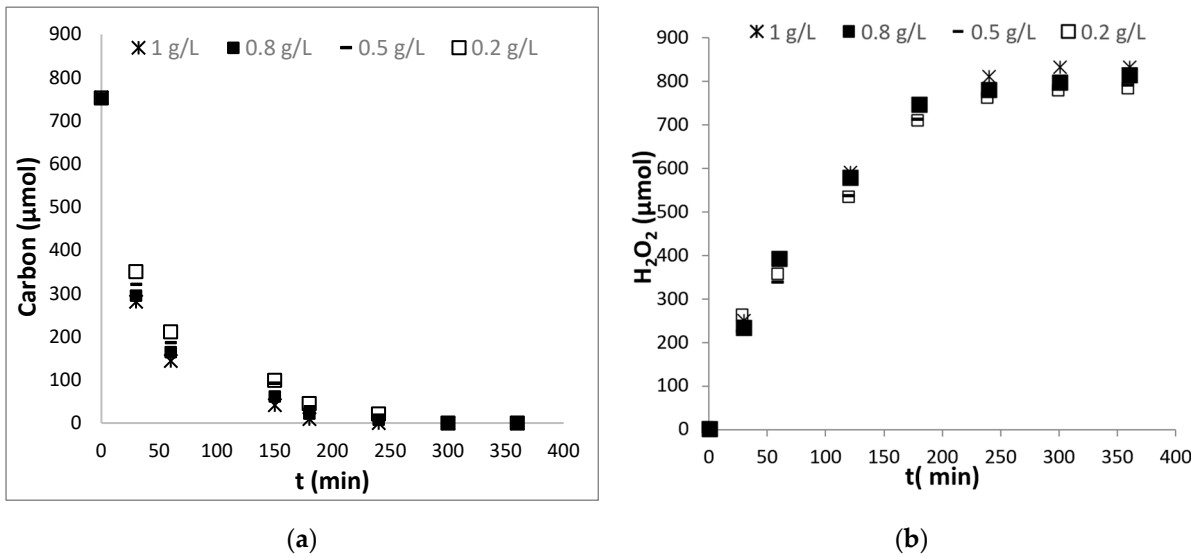

(**a**)         (**b**)

**Figure 6.** Effect of ZnO concentration (0.2 g/L, 0.5g/L, 0.8 g/L, and 1 g/L) on the photocatalytic degradation of phenol (**a**), and the evolution of $H_2O_2$ formed as a function as irradiation time (**b**).

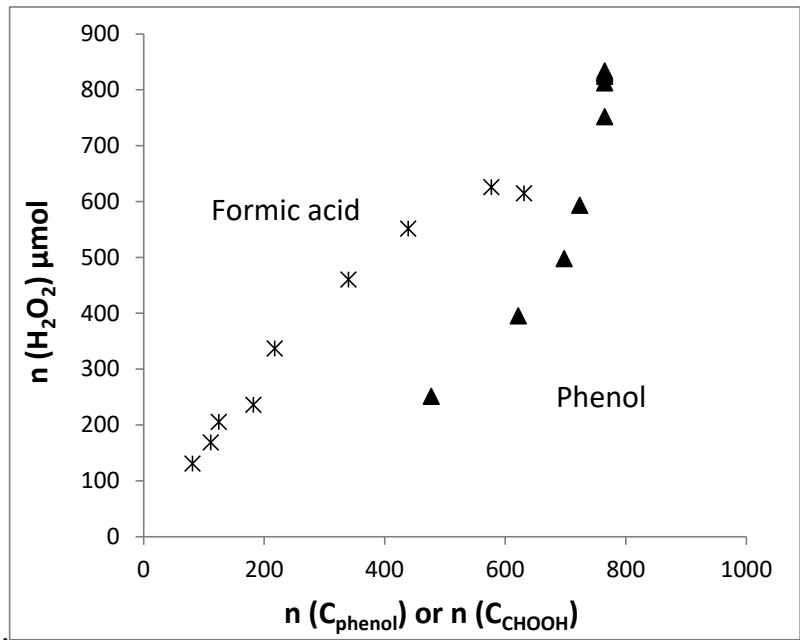

**Figure 7.** Moles of $H_2O_2$ formed as a function of the number of moles of carbon of HCOOH and phenol degraded.

After the complete disappearance of phenol (766 μmol), the $H_2O_2$ formation continues to evolve suggesting that some intermediate products are present and at the origin of the formation of $H_2O_2$. On the contrary, in the case of formic acid, a pseudo plateau begins to appear showing the total disappearance of formic acid in the solution. To be able to compare the number of $H_2O_2$ formed to the amount of carbon degraded for these two pollutants, the intermediate products and the Total Organic Carbon (TOC) content remaining in the solution during phenol degradation was reported in Figure 8a. The intermediate products formed during the degradation of phenol are initially aromatic compounds, hydroquinone (HQ), benzoquinone (BQ), and catechol (CAT) [53–55]. Then, some carboxylic acids are

formed similar to those observed in the case of $TiO_2$ [56] but not quantified in this study. Figure 8b represents the number of $H_2O_2$ formed as a function of Total Organic Carbon (TOC) removed from the HCOOH or phenol.

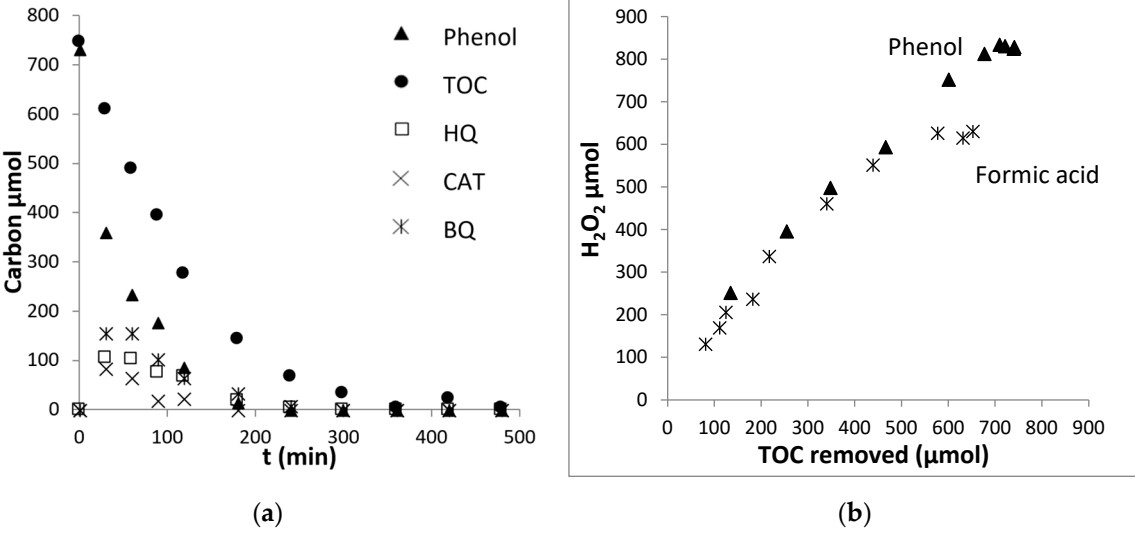

**Figure 8.** Phenol, catechol (CAT), hydroquinone (HQ), benzoquinone (BQ), and Total Organic Carbon (TOC) content as a function of irradiation time (**a**), and mole of $H_2O_2$ formed as a function of Total Organic Carbon removed (**b**).

The results indicate that the number of moles of $H_2O_2$ formed per number of carbon atoms removed is similar to the degradation of these two molecules (formic acid and phenol). After removing the amount of $H_2O_2$ formed in the absence of a pollutant, we found that for 1 mole of carbon removed about 1 mole of $H_2O_2$ is formed along the reaction for these two pollutants.

### 2.3.2. Influence of the Nature of ZnO Samples on $H_2O_2$ Formation

After investigating the formation of $H_2O_2$ in the presence of ZnO-Com and showing that the formation of $H_2O_2$ seems to be correlated with the disappearance of the number of carbon during the degradation of FA and phenol with a factor of about 1 along the degradation, we investigated the behavior of ZnA4T, ZnH4T, and ZnP4T concerning the formation of $H_2O_2$ and the disappearance of carbon during the degradation of FA and Ph. The formation of $H_2O_2$ and the disappearance of FA and Ph in the presence of different ZnO samples are reported in Figure 9. With whichever ZnO samples, $H_2O_2$ is detected during the degradation of FA and Ph. In the absence of a pollutant, the amount of $H_2O_2$ initially increases and reaches a plateau of around one hundred µmol. However, it is difficult, in the absence of a pollutant, to determine the difference in $H_2O_2$ formation between the different catalysts although it seems that less $H_2O_2$ was formed with the sample ZnP4T.

We noticed that the higher disappearance of FA and Ph, the greater the formation of $H_2O_2$ as previously observed with ZnO-Com. So, it means that the formation of $H_2O_2$ depends on the same physicochemical parameter as those influencing the photocatalytic properties. The results are in agreement with the EPR results of ZnO-com in the presence of phenol (Figure S6a), which explains its much higher activity. In the literature, Jang et al. [28] found that the highest activity for $H_2O_2$ generation is obtained with an exposed (001) polar face and depends on the morphology of the crystal. However, in our case, the different ZnO samples have similar morphologies and the same exposed face and cannot explain the difference observed. In agreement with our previous publication [57], the differences observed can be explained by considering the harmful effect of defects for photocatalysis. The defects decrease from ZnP4T to ZnH4T and ZnO-com. In the case of ZnA4T, the higher amount of $H_2O_2$ observed can be explained by considering its large surface area.

Moreover, whatever the ZnO sample, the ratio between the number of $H_2O_2$ formed and the number of carbon atoms removed is around 1, in the presence of FA and Ph (Figure 10), as previously obtained for ZnO-Com. It is also interesting to note that the highest photocatalytic performance and the highest $H_2O_2$ generation is obtained with the ZnO samples with less oxygen vacancies, which is in agreement with our previous results [57].

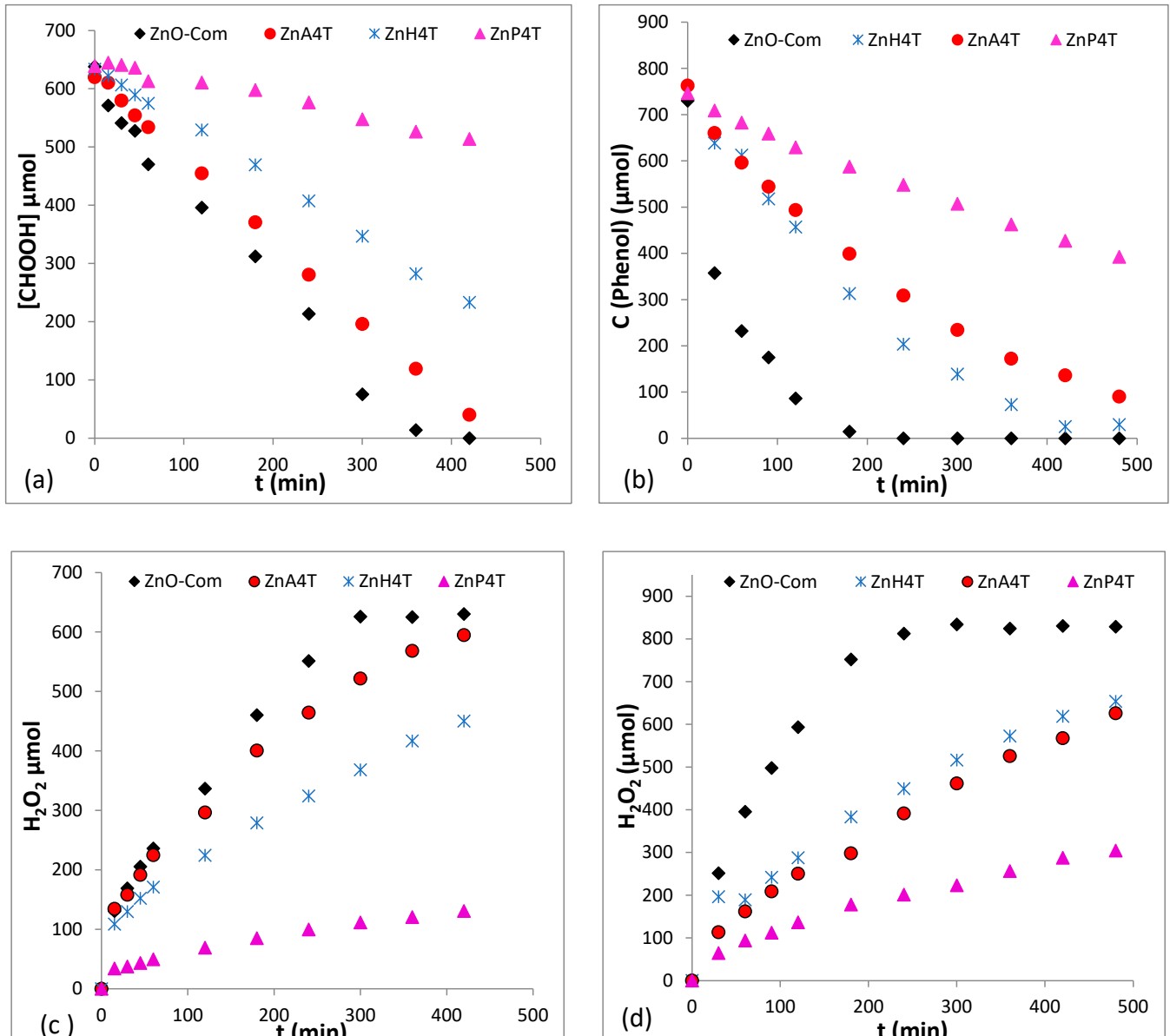

**Figure 9.** Disappearance of FA (**a**) and Ph (**b**) as a function of irradiation time for different ZnO samples, and formation of $H_2O_2$ during FA (**c**), and Ph (**d**) degradation.

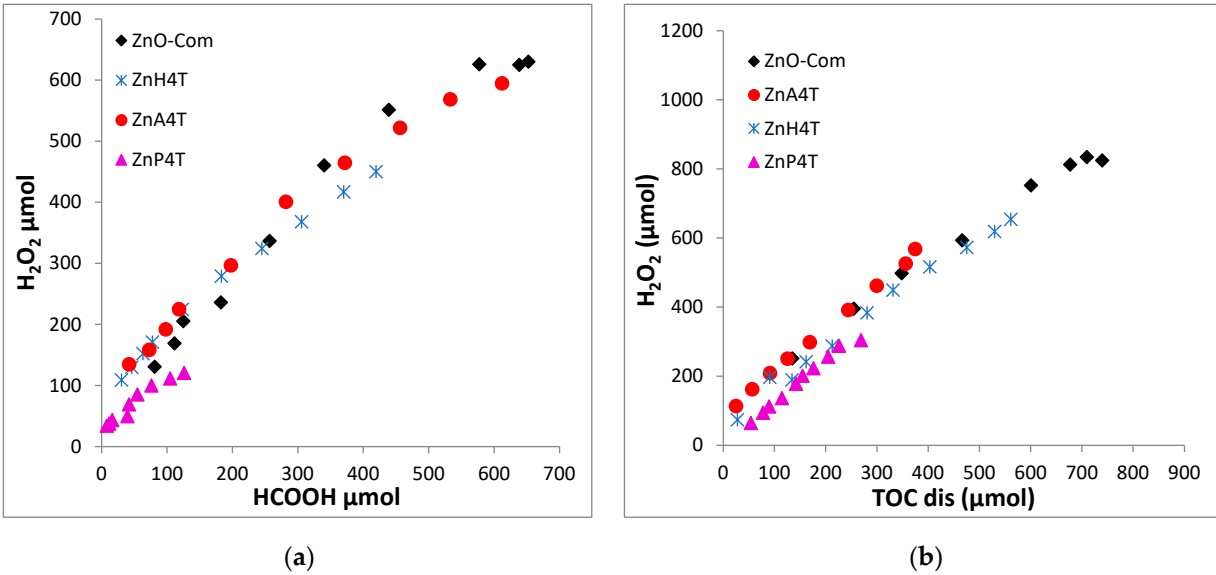

(**a**)                                                          (**b**)

**Figure 10.** Moles of $H_2O_2$ formed as a function of moles of HCOOH (**a**) and phenol (**b**) degraded.

### 2.4. Photocatalytic Decomposition of $H_2O_2$

The production of hydrogen peroxide shown in Section 2.3 has been studied in the presence of various pollutants (FA and Ph) and the impact of using different ZnO was also investigated. For this section, only phenol is considered. Taking into account the different behaviors of ZnO and $TiO_2$ anatase and $TiO_2$ rutile toward the formation and the decomposition of $H_2O_2$ [27], we have mixed 50% of ZnO Com with 50% of $TiO_2$ anatase (and rutile) to determine the impact of both of these $TiO_2$ phases on the formation of $H_2O_2$ generated in the presence of ZnO and on the formation of the first intermediate products, hydroquinone, catechol, and benzoquinone. The formation of $H_2O_2$ and the first intermediate products (hydroquinone (HQ), benzoquinone (BQ), and catechol (CAT)) formed during the degradation of phenol in the presence of ZnO alone or mixed with $TiO_2$ anatase and $TiO_2$ rutile are presented in Figure 11a,b.

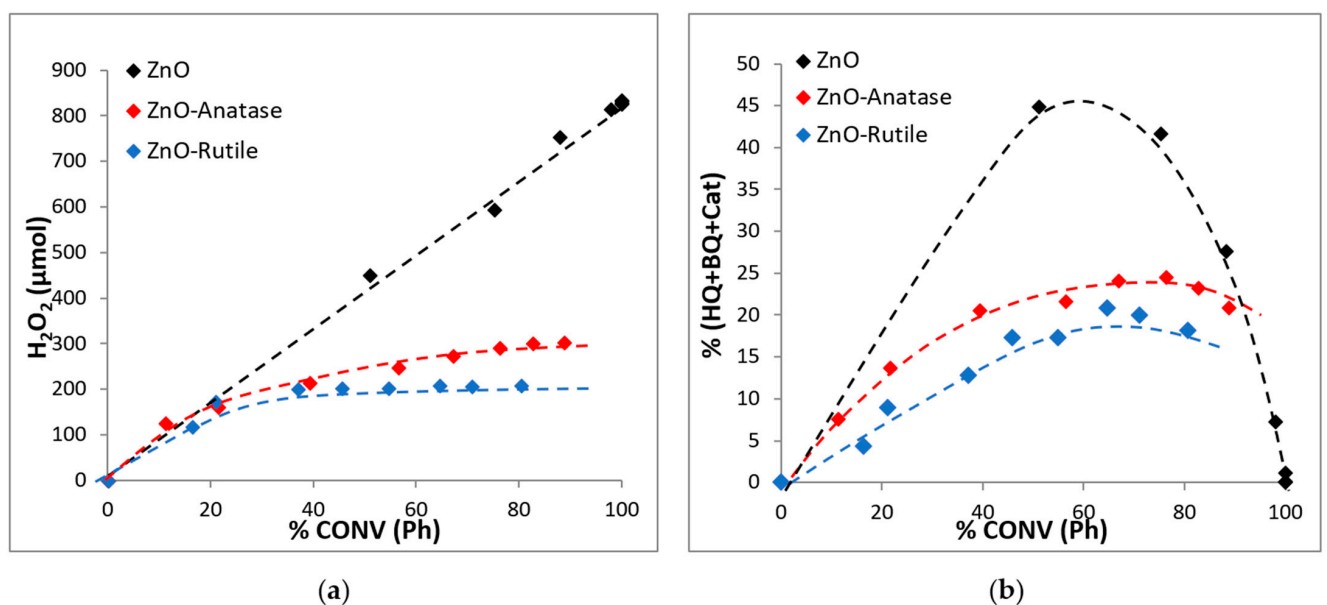

(**a**)                                                          (**b**)

**Figure 11.** Evolution of $H_2O_2$ formed (**a**), and yield into HQ, BQ, and CAT (**b**) as a function of phenol conversion in the presence of ZnO and a mixed ZnO-anatase (50/50), ZnO-rutile (50/50).

On the composites ZnO-TiO$_2$ (anatase) and ZnO-TiO$_2$ (rutile), a decrease in the amount of H$_2$O$_2$ generated compared to ZnO was noticed, which is slightly more important in the presence of ZnO-TiO$_2$ (rutile). The smaller formation of H$_2$O$_2$ in the presence of composites is due to its decomposition on TiO$_2$ and the difference observed in the concentration of H$_2$O$_2$ generated with both of the composites is in good agreement with our previous work indicating the more important decomposition of H$_2$O$_2$ in the presence of the rutile phase [26]. Moreover, in Figure 11b it can be observed that at the same conversion of phenol, the yield of phenolic intermediates in the presence of the two composites was lower compared to this one obtained in the absence of TiO$_2$. These results confirm the interaction of H$_2$O$_2$ generated with the accessible sites of TiO$_2$ to form additional radicals that can participate in the photocatalytic mechanism and enhance the oxidation of intermediate compounds. On the other hand, the yield of the phenolic intermediate is slightly lower for the ZnO-TiO$_2$ rutile compared to the composite ZnO-TiO$_2$ anatase. This difference could be attributed to the nature of the reactive oxygen species (ROS) derived from the decomposition of H$_2$O$_2$ under UV irradiation on TiO$_2$ anatase and rutile. Some publications reported that °OH radicals are preferentially formed in the decomposition of H$_2$O$_2$ on rutile, whereas the production of O$_2$°$^-$ is favored in the presence of TiO$_2$ anatase [17,25,27].

## 3. Materials and Methods

### 3.1. Photocatalysts

Four ZnO samples were studied: a commercial ZnO from Aldric-labelled ZnO-Com (17 m$^2$/g) and three homemade ZnO samples, ZnA4T prepared using Zinc acetate (Zn(CH$_3$COO)$_2$·2H$_2$O), ZnH4T and ZnP4T prepared from Zinc nitrate (Zn(NO$_3$)$_2$·6H$_2$O) after intermediate preparation of Zn(OH)$_2$ by mixing NaOH (2 M), and Zn(NO$_3$)$_2$·6H$_2$O (1 M) for 2 h to obtain Zn(OH)$_2$, washing the gel obtained several times with distilled water and then dried at room temperature for 24 h.

The preparation of these three ZnO samples are reported below.

The elaboration of ZnA4T:Zn(CH$_3$COO)$_2$·2H$_2$O was precipitated u KOH in methanol with a ratio of 4. During synthesis, magnetic stirring was maintained and the temperature was fixed at 60 °C for 3 h under reflux. The resulting product was separated from a solution by centrifugation, washed with ethanol and water, dried at 60 °C, and finally calcined at 400 °C for 2 h under air flux with a heating rate of 5 °C·min$^{-1}$.

The elaboration of ZnH4T was formed by calcination at 400 °C for the Zn(OH)$_2$ previously obtained. The ZnP4T was elaborated by mixing an aqueous solution of Zn(OH)$_2$ and H$_2$O$_2$ (1M), which is kept under stirring at 70 °C for 2 h. Then, the precipitate was separated and washed several times by centrifugation, then heated at 60 °C overnight. The obtained ZnO$_2$ was further calcinated at 400 °C for 2 h to obtain ZnP4T.

Two commercial titanium dioxide catalysts were also used, a rutile phase HPX-400C named R100 (85 m$^2$/g) and an anatase HPX-200/v2 named A100 (96 m$^2$/g)) from Crystal (Thann, France). The obtained 50 wt% ZnO—50 wt% anatase (ZA) and 50 wt% ZnO—50 wt% rutile (ZR) were prepared through a mechanical mixing method using an agate mortar. The resulting powder was kept under incipient wetness stirring and, finally, dried overnight at 60 °C.

### 3.2. Characterization

The as-prepared photocatalysts were characterized through X-ray Diffraction (XRD) using an A25 Bruker D8 Advance diffractometer (Billerica, MA, USA) with Cu-K$\alpha$ radiation ($\lambda$ = 0.15406 nm) at 40 KV, and a scanning range between 2$\theta$ = 4–80° with a scan rate of 0.02°/s. The UV-visible diffuse reflectance considering the Kubelka–Munk method provided the direct bandgap transition for ZnO: ($\alpha$hv)$^2$ versus (hv) where $\alpha$ is the absorption coefficient and hv is the photon energy. The BET surface areas were measured on an ASAP 2020 instrument using nitrogen physisorption at 77 K. The catalysts were degassed at 200 °C for 3 h under vacuum. The Raman experiments were performed with a spectral resolution of 4 cm$^{-1}$ and an Ar+ ion laser at 514 nm (Horiba Jobin Yvon LabRAM-HR equipment,

Palaiseau, France). A CCD detector cooled at $-75\ ^{\circ}$C was used. The morphology of the samples was carried out using Transmission Electron Microscopy (TEM) using a JEOL 2010 microscope (Tokyo, Japan) with 200 KV and the particle size distribution was determined using Image J (Fiji 2.0.0-rc-68/1.53t). The EPR assays were all carried out at room temperature using a Bruker E500 spectrometer operating at X-band (9.34 GHz), sensitive cavity, and with 100 kHz modulation frequency. The instrument settings were as follows: microwave power; 22 mW; modulation amplitude; 1 G. The hyperfine coupling constants (a and g values) were obtained with a simulation of experimental spectra using easyspin (Matlab 2016b (Natick, MA, 01760 USA), Easyspin (easyspin-5.2.33)). The aqueous solutions of a spin trapping DMPO (5,5-Dimethyl-1-Pyrroline-N-Oxide, TCI chemicals) were prepared in capillary tubes. The irradiation (0.5–20 mn), Thorlab LED365 nm (Newton, NJ, USA), was directly performed in the EPR cavity while the spectrum was recording.

### 3.3. Photocatalytic Experiments and Analytical Procedure

For photocatalytic degradation tests, a 1 L Pyrex photo-reactor thermostated by water external circulation was employed. The formic acid (99% pure) and phenol (99% pure) were supplied, respectively, from Across Organics. For all the experiments, the concentration of the photocatalyst was set at 1 g·L$^{-1}$; 600 mg of catalyst were added to 600 mL of a solution of formic acid (50 ppm) or phenol (20 ppm) with bottom magnetic stirring for 30 min in the dark to reach equilibrium. A pre-heated UV-A T8 8W diving lamp (Vilber Lourmat, Collégien, France) was positioned in a quartz tube in the middle of the reactor, providing a 2.4 mW/cm$^2$ irradiation; 1 mL of the solution was sampled and filtrated on an MILLEX HVLP 0.45 µm hydrophilic filter (Millipore, Burlington, MA, USA) for the HPLC analysis and $H_2O_2$ measurements.

For analytical characterization, high-performance liquid chromatography (HPLC) VARIAN PROSTAR (Agilent Technologies, Santa Clara, CA, USA) with an automated sampler was employed for formic acid analysis. An $H_2SO_4$ ($5 \times 10^{-3}$ mol·L$^{-1}$) mobile phase with a flow rate of 0.7 mL min$^{-1}$ was used, equipped with a Coregel-87H3 column (300 mm $\times$ 7.8 mm—Concise Separations). For detection, a Prostar325 UV-Vis module was set at 210 nm. To follow the phenol disappearance and the aromatic intermediates, a 1290 Infinity HPLC system (Agilent Technologies, Santa Clara, CA, USA) equipped with a Nucleosil 250 $\times$ 4.6 mm 5 µm C18 column (Macherey-Nagel, Düren, Germany) was employed. The mobile phase was composed of 20% methanol and 80% of aqueous $H_3PO_4$ (1 mM) solution. A PDA detector was used at 210 nm wavelength. All of the HPLC analysis was performed at 40 $^{\circ}$C.

The Total Organic Carbon (TOC) of phenol was quantified during photocatalytic tests at different times using a Shimadzu model TOC-VSCH with total organic carbon and equipped with an autosampler. The detection limit of the TOC analyzer is 0.5 mg·L$^{-1}$ and the quantification limit is 1 mg·L$^{-1}$.

The hydrogen peroxide measurements were performed on a 6850 UV-Visible spectrometer from Jenway (Cole Parmer, Vernon Hills, IL, USA). For analysis, 500 µL of the solution was sampled and filtrated then mixed with 200 µL of ammonium molybdate ($(NH_4)_2MoO_4$, 0.01 M), 300 µL of $H_2SO_4$ (1 M), 2 mL of potassium iodide (KI, 0.1 M), and 7 mL of water. After being shaken, the solution rests 10 min before being measured at 361 nm.

## 4. Conclusions

The elaboration of ZnO samples from different precursors, zinc acetate, zinc hydroxide, and zinc peroxide were carried out, characterized by RAMAN and EPR and their photocatalytic performance were evaluated using two model molecules, formic acid (FA) and phenol (Ph).

We showed that the most efficient photocatalyst in the degradation of FA or Ph generates the most important amount of $H_2O_2$ correlated to the less important oxygen vacancies. The highest amount of $H_2O_2$ was recorded with commercial ZnO presenting the most important activity for the degradation of FA and Ph, while the sample from

$ZnO_2$ (ZnP4T) generated the lowest amount compared to ZnA4T and ZnH4T. Furthermore, our results show that the number of moles of $H_2O_2$ formed per number of carbon atoms removed during the degradation of FA and Ph was similar. In all cases and regardless of the precursor or the pollutant, a correlation was found between the amount of $H_2O_2$ generated and the amount of carbon removed by a factor of the order of 1.

We also proved that the presence of $TiO_2$ decreases the amount of $H_2O_2$ generated with ZnO. This reduction is due to the decomposition of $H_2O_2$ on $TiO_2$, which is accompanied by an improvement of the degradation of the first phenolic intermediates, more significant in the presence of rutile $TiO_2$. These results are in good agreement with the formation of $^\circ OH$ and $O_2{}^{\circ-}$ radicals in the presence of rutile $TiO_2$ and anatase, respectively, and highlight the important role of $^\circ OH$ in the photocatalytic process.

**Supplementary Materials:** The following supporting information can be downloaded at: https://www.mdpi.com/article/10.3390/catal12111445/s1, Figure S1: X-ray diffraction (XRD) patterns of elaborated precursors $Zn(OH)_2$ and $ZnO_2$; Figure S2: X-ray diffraction patterns of commercial $TiO_2$ (P25, anatase, and rutile); Figure S3: Bandgap determination for $TiO_2$ samples, commercial ZnO as a reference, and composite (ZnO-Anatase and ZnO-Rutile); Figure S4: EPR spectra (Experimental and simulation) of the different ZnO; Figure S5: EPR spectra of ZnO photocatalysts in presence of DMPO-Formic acid (a) and DMPO-Phenol (b). (*: organic species; @: $OH^\circ$ radicals); Figure S6: EPR spectra (Experimental and simulation) of DMPO-Phenol in the presence of commercial ZnO (a) and ZnA4T (b); Figure S7: Calibration curve of $OH^\circ$ radicals obtained with DMPO.

**Author Contributions:** Conceptualization, N.M. and C.G.; methodology, N.M. and C.G.; validation, C.G. and S.P.; formal analysis, N.M., F.D. and L.K.; investigation, N.M., F.D. and L.K; writing—original draft preparation, N.M.; writing—review and editing, C.G., N.J.-R., P.N. and H.B.R.; visualization, N.J.-R., P.N., H.B.R. and A.B.H.A.; supervision, C.G. and P.N.; project administration, C.G. and N.M.; funding acquisition, H.B.R., A.B.H.A. and P.N. All authors have read and agreed to the published version of the manuscript.

**Funding:** This research received no external funding.

**Data Availability Statement:** Not applicable.

**Acknowledgments:** The authors gratefully acknowledge the financial support from the Tunisian Ministry of Higher Education and Scientific Research, the University Lyon 1, ENS Lyon, and the CNRS for hosting me.

**Conflicts of Interest:** The authors declare no conflict of interest.

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
