# Peer review of "Correlation between Photocatalytic Properties of ZnO and Generation of Hydrogen Peroxide—Impact of Composite ZnO/TiO2 Rutile and Anatase"

_catalysts, doi:10.3390/catal12111445_

Round 1

Reviewer 1 Report

In this paper, the production of hydrogen peroxide on ZnO and the photocatalytic performance are studied, and the results are interesting. However, careful examination shows that the paper did not present enough in-depth finding and data analysis. Some explanations need to be further discussed and some details should be corrected.

1.     Throughout the manuscript, the writing of subscripts and superscripts should be uniform.

2.     In addition to ESR analysis, quantitative analysis of radical intermediate should better be provided.

3.     It is recommended to include time-resolved PL (TRPL) measurements, as it can provide intrinsic insights into the carrier lifetime and the material's electron distribution.

4.     The specifications of the figures should be uniform, such as Fig. 2.

Author Response

Thank you for your comments. The answers of each comment have been made. You can find them in attached file. In particular, a calibration curve of °OH radical was done and given in new supplementary documents. Their concentrations were introduced in new table2 of revised manuscript. 

Reviewer 2 Report

The Authors proposed an interesting topic for research. The abstract contains relevant information about the scope of research and results obtained. The paper is well organized and the data are well presented. The paper is publishable, but a minior revision is required before the manuscript acceptance.

Questions and comments:

1.      Relevant information and results are missing from the executive summary.

2.      In the introduction, please specify a clear purpose and novelty of the work.

3.      Line 97-105. The paragraph should be deleted.

4.      Chapter 3.1. Please organize the information on the synthesis of catalysts (e.g. how many catalysts were obtained).

5.      Correctly write down the formulas of chemical substances (stoichiometric indices).

6.      Line 107-108. This paragraph is called "hanging text". Please remove or move.

7.      The symbol for the radical should always be next to the oxygen atom, not the hydrogen atom.

8.      Chapter 2.3. Is this paragraph an overview of the results? The text should be removed.

9.      Were the experiments repeated and the results presented are the average, or were they performed only once?

10.  Figure 11. Is this graph supposed to look like this?

11.  In my opinion, the conclusions are too general.

Author Response

Dear Reviewer, Thank you for all your remarks. You'll find the answer to all your comments in attached file. In particulat we re-written the last part of our introduction to clearly show the novelty of our work and we reorganized the information on the synthesis of catalysts as suggested

Reviewer 3 Report

       In this paper, the authors synthesized different ZnO from different precursors. Further photocatalytic degradation experiments showed that the prepared photocatalysts were able to produce the hydrogen peroxide under UV light irradiation. I would like to see published in Catalysts once the following issues are addressed.

1.     The authors made a poor choice of keywords in the text and suggested replacing H2O2 and ZnO/TiO2. In addition, since oxygen vacancies in ZnO are crucial for the band gap and the final H2O2 yield, it is suggested to include "oxygen vacancies" as a keyword.

2.     Line 97-105 seems to me to be more like a guide than an introduction to the article.

3.     Figure 5 is not clear, please upload high-resolution images

4.     It is suggested that the authors replace the black-and-white images in Figures 4-10 with color-dotted line drawings so that the reader gains a better understanding of the results of the catalytic experiments.

5.     Figure 11(b) shows a large deviation, suggesting the author to revise and re-upload the figure

6.     Authors should state their full name when the abbreviation is first time used, no matter in the abstract or in the text.

7.     Authors should double-check their manuscripts before submitting a revision, there are so many errors. E.g. line 43, “(O2°)”; line47,(H2O2)”; etc…

8.     Several additional articles on TiO2 photocatalysis modified by other strategies are suggested. (e.g., 10.1016/j.apcatb.2016.03.021, 10.3390/nano9030391, 10.1016/j.apcatb.2017.03.077).

Author Response

Dear reviewer, thank you for your comments. You'll find the answers to all your remarks in the attached file given below

Round 2

Reviewer 1 Report

The authors have addressed the issues raised basically, so I think the manuscript could be considered for publication.

Author Response

We corrected a lot of sentences. The corrections are presented in revision mode in the attached file.

We hope now it is OK

Reviewer 2 Report

I recommend the article in this form for publication.

Author Response

The new revised manuscript with correction marked is in attached file

Reviewer 3 Report

The author made changes in this revised manuscript in response to my suggestions, but I think the author did not revise the manuscript carefully enough that the revised manuscript still contains a large number of expression errors! Again, the authors should double-check their manuscript before submission. 

1. line 97-99, another "copy-paste" error;

2. Figure 1. 

3. line 180, (3,01 eV)

4. line 233 H202

5. The description in Figure 5 does not match the main text

6. Photoluminescence performance can verify the separation and transport ability of photogenerated carriers, which is crucial for understanding the improvement of photocatalytic performance. It needs to be supplemented by the authors.

Author Response

Thank you very much for your careful reading. We have rechecked the manuscript and corrected the various points you have noted, the copy-paste line 97-99 is now deleted; Figure 1 corrected, now in line 180 it is 3.01 eV); on line 233 it is H2O2) and we have modified figure 5 (the value is 30 ppm and not 50 ppm). We have also modified some sentences as can be seen in the attached file where the modifications are given in correction mode.

Concerning photoluminescence analysis, we agree with the reviewer, photoluminescence will be very interesting and can complete all the characterizations already done. However, we did not do this type of analysis due to the availability of the device. In the future we will performed photoluminescence. Thank you again for your proposition.

The new revised manuscript with corrections marked in red is given in attached file.
